# Investigating Tacrolimus Disposition in Paediatric Patients with a Physiologically Based Pharmacokinetic Model Incorporating CYP3A4 Ontogeny, Mechanistic Absorption and Red Blood Cell Binding

**DOI:** 10.3390/pharmaceutics15092231

**Published:** 2023-08-29

**Authors:** Matthias Van der Veken, Joachim Brouwers, Agustos Cetin Ozbey, Kenichi Umehara, Cordula Stillhart, Noël Knops, Patrick Augustijns, Neil John Parrott

**Affiliations:** 1Department of Pharmaceutical and Pharmacological Sciences, KU Leuven, 3000 Leuven, Belgium; matthias.vanderveken@kuleuven.be (M.V.d.V.); joachim.brouwers@kuleuven.be (J.B.); patrick.augustijns@kuleuven.be (P.A.); 2Pharmaceutical Sciences, Roche Pharma Research and Early Development, Roche Innovation Centre Basel, 4070 Basel, Switzerland; agustos.ozbey@roche.com (A.C.O.); kenichi.umehara@roche.com (K.U.); 3Pharmaceutical R&D, F. Hoffmann-La Roche Ltd., 4070 Basel, Switzerland; cordula.stillhart@roche.com; 4Laboratory for Pediatrics, Department of Development & Regeneration, KU Leuven, O&N3, Bus 817, 3000 Leuven, Belgium; noelknops@telenet.be; 5Department of Pediatrics, Groene Hart Ziekenhuis, 2803 Gouda, The Netherlands

**Keywords:** tacrolimus, PBPK modelling, absorption, paediatrics, ontogeny

## Abstract

Tacrolimus is a crucial immunosuppressant for organ transplant patients, requiring therapeutic drug monitoring due to its variable exposure after oral intake. Physiologically based pharmacokinetic (PBPK) modelling has provided insights into tacrolimus disposition in adults but has limited application in paediatrics. This study investigated age dependency in tacrolimus exposure at the levels of absorption, metabolism, and distribution. Based on the literature data, a PBPK model was developed to predict tacrolimus exposure in adults after intravenous and oral administration. This model was then extrapolated to the paediatric population, using a unique reference dataset of kidney transplant patients. Selecting adequate ontogeny profiles for hepatic and intestinal CYP3A4 appeared critical to using the model in children. The best model performance was achieved by using the Upreti ontogeny in both the liver and intestines. To mechanistically evaluate the impact of absorption on tacrolimus exposure, biorelevant in vitro solubility and dissolution data were obtained. A relatively fast and complete release of tacrolimus from its amorphous formulation was observed when mimicking adult or paediatric dissolution conditions (dose, fluid volume). In both the adult and paediatric PBPK models, the in vitro dissolution profiles could be adequately substituted by diffusion-layer-based dissolution modelling. At the level of distribution, sensitivity analysis suggested that differences in blood plasma partitioning of tacrolimus may contribute to the variability in exposure in paediatric patients.

## 1. Introduction

Immunosuppression is a cornerstone in the maintenance therapy of organ transplant patients. The macrolide lactone tacrolimus is a potent immunosuppressing agent and one of the key drugs available for such therapy. Tacrolimus has a relatively high lipophilicity and is considered a biopharmaceutics classification system (BCS) class 2 compound due to its low crystalline solubility and high permeability. To overcome absorption issues related to its low solubility, tacrolimus is commercially formulated for oral administration as an amorphous solid dispersion with the innovator product Prograf^®^. Additionally, it is characterised by a nonlinear blood plasma partitioning into the red blood cells and it is highly metabolized by CYP3A [1,2]. Renal clearance of tacrolimus is less than 1% and as such negligible [3]. Since oral dosing of tacrolimus is challenged by a high variability in exposure, therapeutic drug monitoring is routine clinical practice, not only in the adult population, but also in specific populations such as children. As transplantation can happen at an early age and a lifetime of immunosuppressive therapy may be required, large PK datasets for tacrolimus, covering a wide age range, are available [4]. This makes tacrolimus an interesting candidate for analysis with physiology-based pharmacokinetic (PBPK) and population pharmacokinetic (PopPK) modelling. 

PBPK modelling is a valuable tool in drug development and its importance has increased rapidly over the last decades [5]. This mechanistic modelling approach combines physicochemical input data for the drug of interest with physiological data for the subject or population of interest to predict drug exposure profiles and PK parameters. Due to its mechanistic character, PBPK modelling can also be used to investigate the effect of physiological variations and the impact of extrinsic factors such as drug–drug interactions. PBPK modelling is especially helpful in the development of drugs tailored to the paediatric population, by assessing the impact of paediatric physiology on drug performance or assisting in dose setting [6]. 

For tacrolimus, PBPK modelling has been helpful in the assessment of drug–drug interactions [7,8,9,10,11], the influence of drug substance properties [12,13,14] and the causes of exposure variability in adults [15,16,17,18]. However, the paediatric population remains under-investigated, with only Emoto et al. and Zhao et al. [7,16] including this specific population and only Emoto et al. exploring exposure variability. Emoto et al. used sensitivity analyses to study the effect of key covariates such as hepatic CYP3A4 expression and ontogeny, CYP3A5 genotype, body weight, haematocrit, serum albumin, and creatinine levels [16]. They showed that ontogenetic and disease-related changes in hepatic CYP3A4 abundance were the most influential factors [16]. 

In addition to PBPK, PopPK analysis has also been helpful in exploring the variability in tacrolimus PK. The literature data was extensively reviewed by Kirubakaran et al., [19] and CYP3A5 genotyping and haematocrit are most often identified as the major influential cofactors. The PopPK literature also reports an age-dependent effect on tacrolimus exposure, which necessitates dosage adjustments [20,21]. By combining in vitro experiments with PBPK modelling strategies, the present paper aimed to further unravel the underlying causes of the age dependency of tacrolimus pharmacokinetics from an absorption, metabolism, and distribution perspective [20,21]. Paediatric model performance was assessed using an in-house dataset of paediatric kidney transplant PK profiles.

## 2. Materials and Methods

### 2.1. Chemicals Used for In Vitro Experiments

Tacrolimus was procured from Lucerna Chem AG (Lucerne, Switzerland) and the used Prograf^®^ capsule formulation was manufactured by Astellas (Tokyo, Japan). Fasted and fed state simulated intestinal fluid (3F powder) powder was bought from Biorelevant (London, UK). The solvents methanol, acetonitrile, and formic acid (99%) were purchased from Biosolve (Valkenswaard, The Netherlands). Purified water was produced using a Purelab^®^ Flex water system from Veolia (Paris, France). Sodium acetate was obtained from VWR chemicals (Radnor, PA, USA) and glacial acetic acid from Chem-Lab (Zedelgem, Belgium). Both products were used to make the acetate buffer at pH 4.5 according to the United States Pharmacopoeia (USP). 

### 2.2. In Vitro Experiments

#### 2.2.1. Solubility

The solubility of tacrolimus, starting from crystalline powder or an amorphous formulation (Prograf^®^ capsules), was measured in acetate buffer pH 4.5, simulated gastric fluid (SGF) pH 2, half concentrated FaSSIF (½ FaSSIF), FaSSIF, and FeSSIF. SGF medium was prepared by adding 2 g/L of sodium chloride to water and adjusting to pH 2 using 1M HCl. FaSSIF and FeSSIF v1 were prepared according to Biorelevant.com using FaSSIF/FeSSIF/FaSSGF powder. The same procedure was used for ½ FaSSIF except that half the amount of FaSSIF powder was added to the buffer. The full composition of these media is available in the Appendix A. 

Solubility testing was performed by adding an excess amount of crystalline tacrolimus or Prograf formulation to 2.5 mL of each medium. Samples were placed in an end-over-end stirrer set a 5 rpm and kept at 37 °C for the biorelevant media and 25 °C for the acetate buffer. To assess the amorphous solubility (tacrolimus formulation), samples were taken after 0.5, 1, and 4 h. To assess the crystalline solubility, samples were taken after 1, 4, and 24 h. Samples were centrifuged at 20,817 G for 90 s to separate undissolved material. The supernatant was then directly diluted with the mobile phase of the analytical method (see Section 2.2.3). All experiments were performed in triplicate. 

#### 2.2.2. Dissolution

Dissolution experiments were performed in FaSSIF medium using the tacrolimus formulation (Prograf^®^ 5 mg capsules). To increase the biorelevance of the dissolution setup, custom dissolution conditions for all paediatric subpopulations and the adult population were applied. To select appropriate doses, the recommended starting dose for children of 0.15 mg/kg was selected both for adults and children. The weight of an average child for each subpopulation was deduced from the ideal body weight curves provided by the Centre for Disease Control (CDC). For adults, a standard weight of 70 kg was used. The fluid volume used in the in vitro setup was set at 50 mL for adults, based on a measured average small intestinal fluid volume (SIFV) of 43 mL [22]. For the paediatric population, in vitro fluid volumes were chosen based on the average SIFV measured by Van der Veken et al. [23]. A summary of the dissolution conditions can be found in Table 1. 

The dissolution tests were conducted in a small-scale dissolution setup; doses and fluid volumes were scaled down to a constant volume of 15 mL to ensure similar fluid dynamics independent of the tested condition. However, the original dose/volume ratio was maintained. The corresponding amount of Prograf granules was weighed into a 20 mL screw-cap glass vial; at t = 0, 15 mL of each medium was added to the formulation and the vials were placed in an end-over-end rotator at 5 rpm and at 37 °C. Sampling was performed every 10 min for 60 min by taking 300 µL samples. The samples were centrifuged at 20,817 G for 90 s followed by immediate dilution of the clear supernatants with the starting mobile phase of the analytical method. All dissolution tests were performed in triplicate. 

#### 2.2.3. Tacrolimus Analysis

For practical reasons, tacrolimus concentrations were measured using either a UPLC-MS and a UPLC-UV method. A summary of the analytical method parameters is available in Table 2. A more in-depth description of the used UPLC methods is available in the Appendix A. 

### 2.3. PBPK Model Building

#### 2.3.1. Software Tools

PBPK models were developed using two software platforms: SimCYP version 20 release 1 (Certara, Princeton, NJ, USA) and Gastroplus 9.8.3 (Simulations Plus, Lancaster, CA, USA). Additionally, the SimCYP SIVA toolkit was applied to estimate bile micelle partitioning coefficients from in vitro solubility data. To digitize the literature data, Webplotdigitizer (Version 4.6, https://automeris.io/WebPlotDigitizer, accessed on 26 June 2023) was used. 

#### 2.3.2. Model Performance

To assess model performance in simulating the reference data, a visual check was supplemented by numerical metrics, namely the fold error (FE) (Equation (1)), average fold error (AFE) (Equation (2)), and absolute average fold error (AAFE) (Equation (3)). The fold error indicates the error of a single simulated value. AFE indicates the overall bias (i.e., an over- or under-prediction) and AAFE indicates the overall precision for multiple predictions. For the paediatric population, the slope of the FE as a function of age was assessed to determine age-dependent effects.
(1)$$\mathrm{FE}=\frac{\mathrm{Simulated}\;\mathrm{parameter}}{\mathrm{Observed}\;\mathrm{parameter}}$$
(2)$$\mathrm{AFE}=10^{\frac{1}{n}\sum \log\left(\frac{\mathrm{Predicted}}{\mathrm{Observed}}\right)}$$
(3)$$\mathrm{AAFE}=10^{\frac{1}{n}\sum \left|\log\left(\frac{\mathrm{Predicted}}{\mathrm{Observed}}\right)\right|}$$

Throughout the modelling assessment, a 2-fold AAFE was used as threshold for acceptable model performance while a 1.25-fold AAFE indicated good model performance.

These metrics were applied to several PK parameters defined as follows. C_0_ is defined as the last measured or simulated concentration before the next administration. C_max_ is the highest measured concentration in vivo, with the t_max_ being the time at which this concentration is reached. Since only 4–5 time points were available per paediatric patient, it is possible that the exact C_max_ was missed and so the measured C_max_ was compared to the simulated concentration at the t_max_ of the reference data. AUC was calculated using the trapezoidal method for both the simulated and measured concentration–time profiles. The k_e_ was determined using Equation (4) with t_0_ being the time of C_0_ measurement.
(4)$$k_{e}=-\frac{\ln\left(C_{\max}\right)-\ln\left(C_{0}\right)}{t_{\max}-t_{0}}$$

For the paediatric population, the fold error for each predicted concentration was also determined and the average of these fold errors per patient was used to assess how well the overall pharmacokinetic profile was predicted.

#### 2.3.3. SimCYP Model Building

##### Adult Model

To develop the SimCYP model, the approach described by Kuepfer et al. was used [24] (Figure 1). Input parameters were extracted from literature and are reported in Table 3. Data for concentration dependent blood-to-plasma (B/P) partitioning were taken from Jusko et al. [2]. The estimated maximal erythrocyte/plasma ratio was 80. Considering the high distribution into red blood cells, in vivo concentrations of tacrolimus are measured in whole blood and so we considered it essential to include this in the model. For whole-body distribution, a minimal PBPK model was constructed based on the IV and oral data published by Bekersky et al. [25]. The volume of distribution at steady state (V_ss_) was predicted using the Rodgers and Rowland method (Method 2), but this greatly underestimated the observed V_ss_ and so a scalar for all tissue partition coefficients (K_p_ scalar) of 10 was required to achieve a V_ss_ of 9.7 L/kg. Next, the K_in_ value published by Emoto et al. [16] was used while the K_out_, volume of the single adjusting compartment (V_sac_) and K_p_ parameters were fitted based on the clinical data [25] (Figure 1, step 1). The absorption of tacrolimus was simulated using the advanced dissolution, absorption, and metabolism (ADAM) model in SimCYP. The human effective permeability (P_eff_) was estimated from a study performed in rats [26]. For the metabolism, in vitro data obtained using recombinant CYP3A4 and CYP3A5 enzymes as published by Dai et al. were used [1]; this paper characterized two major metabolic pathways which were both included in the model. In line with the available literature PBPK models, the unbound Michaelis constant (K_m,u_) was used as input [1].

**Figure 1 pharmaceutics-15-02231-f001:**
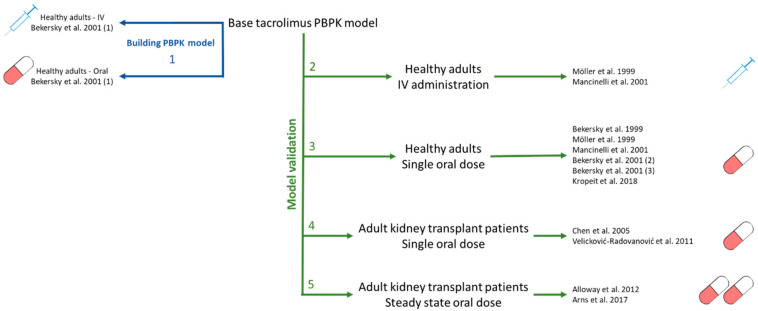
Model development and validation strategy used for the SimCYP adult model building; Bekersky 2001 et al. (1) [25], Möller et al. [3], Mancinelli et al. [27], Bekersky 1999 et al. [28], Bekersky 2001 et al. (2) [29], Bekersky et al. (3) [30], Kropeit et al. [31], Chen et al. [32], Velicković-Radovanović et al. [33], Alloway et al. [34], Arns et al. [35].

**Table 3 pharmaceutics-15-02231-t003:** SimCYP PBPK input parameters.

	Adult Model	References
**Population**	Healthy Volunteers	
**Drug parameters**		
Physicochemical properties		
Molecular weight (g/mol)	804.02	[36]
Log P_o:w_	3.26	[36]
Compound type	neutral	[36]
**Blood binding**		
Concentration-dependent B/P profile		
B_max_ (E:P)	80	[2]
K_D_ (µM)	0.004726	[2]
Fraction unbound in plasma (%)	1.2	[37]
**Absorption**		
ADAM model		
P_eff_ (10^−4^ cm/s)	4.77	[26]
DLM model intrinsic solubility (mg/mL)	0.06257	Aqueous buffer solubility for tacrolimus formulation
Logarithm of bile micelle:buffer partitioning coefficient (log K_m:w_)	4.55	Fitted using SIVA based on solubility data in acetate buffer, ½ FaSSIF, FaSSIF and FeSSIF
Monodispersed particle size distribution radius (µm)	10	Default SimCYP value
**Distribution**		
Distribution model	Minimal PBPK	
k_in_ (1/h)	0.68	[16]
k_out_ (1/h)	0.12	Fitted based on IV data [25]
V_sac_ (L/kg)	5.5	Fitted based on IV data [25]
K_p_ scalar	10	Fitted based on IV data [25]
Predicted V_ss_ (L/kg)	9.68	Predicted in SimCYP
Prediction method	Method 2	
**Elimination**		
Pathway 1		
CYP3A4		
V_max_ (pmol/min/pmol CYP)	8	[1]
K_m,u_ (µM)	0.21	[1]
CYP3A5		
V_max_ (pmol/min/pmol CYP)	17	[1]
K_m,u_ (µM)	0.21	[1]
Pathway 2		
CYP3A4		
V_max_ (pmol/min/pmol CYP)	0.6	[1]
K_m,u_ (µM)	0.29	[1]
CYP3A5		
V_max_ (pmol/min/pmol CYP)	1.4	[1]
K_m,u_ (µM)	0.35	[1]

B_max_ = maximum E:P ratio, K_D_ = constant of the drug–erythrocyte complex, P_eff_ = effective permeability, K_in_ = rate constant into the single-adjusting compartment, K_out_ = rate constant out of the single-adjusting compartment, V_Sac_ = volume of the single-adjusting compartment, K_P_ scalar = scalar for all tissue partition coefficients, V_SS_ = volume of distribution at steady-state, K_m,u_ = unbound Michaelis–Menten constant.

To validate model performance, data obtained after dosing healthy adults with a single dose of tacrolimus intravenously [3,27] (Figure 1, step 2) and orally [3,25,27,28,29,30,31] (Figure 1, step 3) were simulated, adjusting the system parameters to match the demographics of actual study participants. For all oral administrations, the diffusion layer model (DLM) was used to simulate tacrolimus dissolution. Here, the intrinsic solubility was set to the measured solubility starting from the Prograf capsule formulation in acetate buffer. The bile micelle:buffer partitioning coefficient (K_m:w_) was estimated from the solubility data in acetate buffer, ½ FaSSIF, FaSSIF, and FeSSIF using the SIVA toolkit. The parameter estimation for K_m:w_ used the measured input solubilities with the corresponding bile salt concentrations of the media. K_m:w_ was estimated for both crystalline and formulated tacrolimus; since values were very comparable, they were averaged to obtain input K_m:w_ for SimCYP. As tacrolimus is a neutral compound, no effect of pH was expected. One study from Möller et al. was exceptional since tacrolimus was administered as a solution; for this case, the dosage form was adapted accordingly.

Next, the model performance was assessed for kidney transplant patients receiving a single dose of tacrolimus based on the data published by Velicković-Radovanović and Chen et al. [32,33] (Figure 1, step 4). Model performance was further tested against the steady-state plasma concentrations published by Alloway et al. and Arns et al. [34,35] (Figure 1, step 5). No physiological adaptations were made to the population to simulate kidney transplant patients except that the weight and age ranges were matched to the reference data.

Dissolution Model Comparison in the Adult Population

The impact of different dissolution models on the PK prediction was assessed by comparing simulations using the DLM to simulations using direct input of measured, biorelevant in vitro dissolution data for the Prograf capsule formulation in FaSSIF. Based on the slope of the last 2 measured timepoints, the in vitro dissolution profile was extrapolated to a release of 93.6% (corresponding to the determined tacrolimus solubility from Prograf in the same FaSSIF medium). The extrapolation indicated that this release would be achieved after 2.24 h.

Precipitation was not investigated for the adult population as a study by Trasi et al. revealed no precipitation within 2 h, which is a reasonable timeframe for absorption of a highly permeable compound such as of tacrolimus [38].

##### Extrapolation to Paediatrics

Paediatric reference dataset for tacrolimus

To assess the paediatric model performance, simulated results were compared to an in-house dataset of paediatric kidney transplant PK profiles, previously the basis for a PopPK analysis [21]. To reduce the potential influence of drug–drug interactions on tacrolimus PK, only data from patients receiving no antiviral medication were considered. The final dataset consisted of 18 children aged between 3 and 18 years with 47 concentration profiles measured on different occasions. Each PK profiling included on average 6 time points distributed over a 12 h period. Based on genotyping, none of the included patients were high metabolizers. As Emoto et al. [16] reported the best model performance when using a liver CYP3A4 ontogeny adapted from Salem et al. [39] (here referred to as the Emoto ontogeny), this ontogeny profile was included in our model. Next, the impact of CYP3A4 ontogeny (Figure 2, step 1–3), absorption (Figure 2, step 4, 5, 8 and 9), and distribution (Figure 2, step 6–7) on the prediction of tacrolimus exposure in children was assessed.

2.Paediatric CYP3A4 and CYP3A5 ontogeny profiles

For the expression of CYP3A4 and CYP3A5, different ontogeny profiles in the liver and intestines are available in SimCYP. For CYP3A4, depending on the selected profile in the liver, a default combination with an intestinal ontogeny is made.

The liver CYP3A4 ontogeny as described by SimCYP profile 1 is based on the data published by Salem et al. [39] and is by default combined with the intestinal ontogeny as described by Johnson et al. [40]. Throughout the rest of this paper, this is referred to as “Salem ontogeny”. The liver CYP3A4 ontogeny as described by SimCYP profile 2 is based on the data published by Upreti et al. [41] and is by default combined with no ontogeny profile in the intestine. Throughout the rest of this paper, this is referred to as “Upreti ontogeny”.

Additionally, Emoto et al. explored a different hepatic CYP3A4 ontogeny to predict tacrolimus exposure based on SimCYP profile 1 [16]. As such, their new liver ontogeny profile is combined with the Johnson et al. intestinal CYP3A4 ontogeny. Throughout the rest of the paper, this is referred to as “Emoto ontogeny”. Lastly, a new ontogeny combination is introduced. Here, the Upreti liver ontogeny profile is used for both hepatic and intestinal CYP3A4 expression. Throughout the rest of this paper, this is referred to as “Upreti liver + intestine ontogeny”. A summary of the different combinations is available in Table 4; the age dependency of all CYP3A4 profiles is visualized in Figure 3. Input parameters for the sigmoidal functions to obtain these ontogeny profiles are available in Appendix A.

For CYP3A5, only 1 default combination is available in SimCYP. For hepatic CYP3A5, no ontogeny profile is used. For intestinal CYP3A5, the profile as described for CYP3A4 by Johnson et al. [40] is used. These ontogeny profiles were left unchanged throughout this paper as none of the included subjects were high metabolizers.

To analyse the impact of CYP3A4 for ontogeny, the model performance using the Emoto ontogeny was compared to the Upreti ontogeny (Figure 2, step 2). Next, the model performance was then compared to the Upreti liver + intestine ontogeny (Figure 2, step 3).

3.Exploring the impact of dissolution rate and bile salt levels on paediatric absorption of tacrolimus

Simulations using the DLM were compared to those obtained when using measured dissolution profiles for different paediatric subpopulations as direct inputs (Figure 2, step 4). This allowed us to assess the impact of reduced gastrointestinal fluid volumes in the paediatric population since duodenal fluid volumes cannot be changed directly in SimCYP version 20.

Next, the impact of reduced bile salt levels in younger children was evaluated using the DLM (Figure 2, step 5). This was achieved by using the measured solubility in different media (acetate buffer, ½ FaSSIF, normal FaSSIF) as the intrinsic solubility input. The K_m:w_ was set to 0 to prevent the solubility-enhancing effect of bile salts from being taken into account twice. Lastly, to determine the effect of the K_m:w_, a sensitivity analysis was run on the K_m:w_ over a range based on the confidence interval of the K_m:w_ obtained in SIVA.

4.Investigating impact of distribution on tacrolimus exposure

The impact of distribution was explored as a potential source of variability in the pharmacokinetics by varying the B_max_ for blood:plasma partitioning (Figure 2, step 6) and the fraction unbound in plasma (Figure 2, step 7).

For blood:plasma partitioning, the amount of tacrolimus partitioning into the red blood cells was varied depending on age according to the B:P_max_-input profile shown in Figure 4, as the observed high blood:plasma partitioning observed for tacrolimus seems to be driven by the binding to target proteins (FK binding protein (FKBP)) inside the red blood cells [42]. As such, it is hypothesized a decreased expression of these FKBP might be possible, thus reducing the amount of drug partitioning in the red blood cells for younger children. The age-dependent effect on exposure as described by Knops et al. was used as a basis for this profile [21]. This effect was hypothetically attributed to the B_max_ by using the adult B_max_ for the 18-year-old (yo) patients and fitting the B_max_ for the 3yo patients. B_max_ values for other ages were interpolated based on the profile published by Knops et al. [21] and displayed below (Figure 4). For the fraction unbound in plasma, the impact of 2-, 3- and 4-fold increases in younger children was assessed using a sensitivity analysis in 3 yo patients (Figure 2, step 7).

#### 2.3.4. IVIVE Sensitivity Analysis on Tacrolimus Fraction Absorbed in Gastroplus

To further investigate the effect of SIFV on the absorption of tacrolimus, a mechanistic absorption model was developed in Gastroplus (Figure 2, step 8) and the sensitivity of the fraction absorbed to SIFV was assessed. This was necessary as the duodenal fluid volume cannot be adjusted for the paediatric population in Simcyp. This sensitivity analysis was performed for both crystalline and formulated tacrolimus by fitting different solubilisation ratios to the experimentally determined solubilities in Gastroplus. For the sensitivity analysis, the baseline, minimum, and maximum values were chosen as the average, lowest, and highest SIFV measured by Van der Veken et al. [23], respectively. Other Gastroplus input parameters were the same as those used in the SimCYP model but no distribution or elimination parameters were necessary; only the fraction absorbed was investigated. A summary of all input parameters for the Gastroplus model and the parameter sensitivity analysis can be found in Table 5.

Additionally, the possible effect of recrystallization from the amorphous formulation was explored (Figure 2, step 9) by using the solubility of crystalline tacrolimus as the solubility of the precipitate. For the dose and SIFV, a high 10.5 mg dose with the average SIFV of 45.2 mL was chosen and a sensitivity analysis for fraction absorbed as a function of the precipitation time was run. For the precipitation time, a minimum of 0.1 s, a baseline of 90 s and a maximum of 900 s were chosen for 40 logarithmically distributed runs.

## 3. Results and Discussion

### 3.1. In Vitro Characterization

#### 3.1.1. Solubility in Biorelevant Media

Tacrolimus solubility starting from crystalline powder and the amorphous Prograf formulation was measured in the biorelevant media SGF, ½ FaSSIF, FaSSIF, and FeSSIF, as well as in acetate buffer pH 4.5. The crystalline equilibrium solubility was measured after 24 h. When using the amorphous formulation, concentrations measured after 30 min and 1 h were similar; a decrease in concentration was observed after 4 h, which is likely due to formation of a less soluble precipitate. The amorphous solubility was therefore measured after 1 h. As can be seen in Figure 5, the amorphous solubility of tacrolimus was at least 7.39-fold higher than its crystalline solubility (SGF solubility). In general, both solubilities increased with an increase in bile salt and phospholipid concentrations in the biorelevant media. All solubility data are available in Appendix A.

Next, these solubility data were used to determine the bile micelle:buffer partitioning coefficient K_m:w_ using the SIVA toolkit. The determined log K_m:w_ values were similar for crystalline (4.52 with a 5–95% confidence interval of 4.1–4.9) and amorphous tacrolimus (4.58 with a 5–95% confidence interval of 3.6–5.5). Given these very similar values, the average K_m:w_ of 4.55 was used in the DLM simulations for both crystalline and amorphous tacrolimus.

#### 3.1.2. Dissolution Profiles

The dissolution rate of Prograf granule formulation was measured in FaSSIF by simulating the dose-to-SIFV ratio in different paediatric subpopulations and in adults. When simulating the paediatric conditions, similar dissolution patterns were observed with complete release within 60 min (Figure 6). In contrast, the adult conditions resulted in a slower release, reaching a fraction dissolved of only 76 ± 3% after 60 min, equivalent to a concentration of 156 ± 3 µg/mL. The slower and incomplete dissolution can be attributed to the much higher dose/volume ratio for adult versus paediatric conditions (0.210 mg/mL for adults vs. ≤0.160 mg/mL for children). The concentration of dissolved tacrolimus after 60 min in adult conditions (156 ± 3 µg/mL) largely exceeds the crystalline solubility (6.78 µg/mL) and approximates the amorphous solubility in FaSSIF (196.6 ± 0.8 µg/mL).

The dissolution behaviour is in line with the literature data from Purohit et al. who determined the effect of crystallinity on dissolution using an aqueous buffer in USP I and II setups and showed a similar fast and extensive dissolution of the amorphous formulation [12].

### 3.2. PBPK Modelling

#### 3.2.1. Adult Population

##### IV and Oral Administration

A PBPK model to predict exposure of tacrolimus in healthy adults was developed based on the input data shown in Table 3 and using the PK data published by Bekersky et al. [25] (Figure 1, step 1, Appendix A). Reference data from Möller et al. and Mancinelli et al. were then used to validate IV model performance [3,27] (Figure 1, step 2). Performance was deemed good based on visual comparison of the simulated exposure to the observed concentrations (Figure 7A and Appendix A). In addition, all observed profiles fell within the 5–95% percentile predicted by SimCYP and both AFE and AAFE were between 1.10 and 1.18 for all PK parameters (Appendix A).

Next, the model was used to simulate tacrolimus exposure after oral administration, using the DLM to model dissolution. Model performance was assessed using reference data from five clinical studies in healthy adults [25,28,29,30,31] (Figure 1, step 3). Model performance for the different references was deemed good based on visual comparison of simulated profiles to observed data and the (A)AFE (Appendix A). For Möller et al., which used a solution administration, model performance was acceptable (Appendix A).

Lastly, model performance was assessed using clinical data from adult kidney transplant patients, both after a single dose and at steady state [32,33,34,35] (Figure 1, step 3 and 4). Model performance was again acceptable with FE between 0.88 and 1.5. An example of predicted versus observed tacrolimus exposure can be found in Figure 7B (single dose) and Figure 7C (steady state).

Looking at the predictions for all reference data combined and using the DLM to simulate dissolution after oral administration, the AFE amounted to 0.91, 1.13, and 1.19 for AUC, C_max_, and k_e_, respectively. The AAFEs amounted to 1.19, 1.20, and 1.20 (Appendix A). The resulting simulated PK profiles, AFE, and AAFE for the different PK parameters are available in the Appendix A).

##### IVIVE to Explore the Impact of Dissolution Models on Tacrolimus Exposure

After model performance was deemed good for IV and oral administrations in adults, the validated model was used to evaluate the impact of drug dissolution on oral tacrolimus exposure. A comparison was made between simulated pharmacokinetics which used (i) the DLM or (ii) the measured biorelevant in vitro dissolution profile for adults in FaSSIF (Figure 6). Möller et al. was not included into this evaluation due to the administration as a solution. Fold errors on the prediction of AUC, C_max_, and k_e_ are visualized for the two dissolution models in Figure 8. The simulated PK profiles and PK parameters for the different reference studies are available in Appendix A.

Based on the simulated results, tacrolimus exposure after oral Prograf administration was well predicted, irrespective of the dissolution model used. AAFEs for the different models and PK parameters were all ≤1.20 (Appendix A). AFEs were between 0.91 and 1.19 for all predicted parameters and dissolution models. Individual simulations of the adult reference data using the different dissolution models are available in Appendix A. Observed and predicted PK parameters using the different dissolution models are available in Appendix A.

#### 3.2.2. Paediatric Population

Having verified the model performance for adults, the system parameters were switched to the SimCYP “Sim-Paediatric” paediatric population. The DLM was retained to predict dissolution and the Emoto ontogeny [16] was used for CYP3A4 and CYP3A5 expression in the liver and intestines (Figure 3, Table 4). As can be seen in Figure 3, a fairly high similarity exists between the Emoto liver CYP3A4 ontogeny and the Upreti liver CYP3A4 ontogeny. Considering that the more recent literature on PBPK modelling of CYP3A substrates in children tends to report a better performance with the Upreti ontogeny [43], we also evaluated model performance when using the Upreti ontogeny. The results for both paediatric model 1 (Emoto ontogeny) and model 2 (Upreti ontogeny) are presented in Table 6. A visualization of the obtained fold errors as a function of age on different PK parameters is available in Figure 9. As previously mentioned, model performance was assessed using the paediatric reference data set consisting of blood concentration profiles for 18 individual children between age 3–18.

In line with the observations of Emoto et al., the model using the Upreti ontogeny performed worse than the model using the Emoto ontogeny (higher AAFE for AUC, C_max_ and C_0_, Table 6 model 1 and 2). Figure 9 illustrates that most individual FE on PK parameters outside the two-fold acceptance range result from the model using the Upreti ontogeny.

It should be noted, however, that the Upreti ontogeny was built based on IV data. As such, intestinal metabolism is less accounted for. When using the default Upreti ontogeny in SimCYP, it only applies to the liver and no intestinal CYP3A4 ontogeny is assumed (fraction of adult expression = 1 independent of age, Figure 3). As tacrolimus is extensively metabolized by the gut (fraction escaping gut metabolism (F_g_) = 0.14–0.26 [16]), intestinal CYP3A4 expression plays a major role in oral exposure. Therefore, an approach which used the same Upreti ontogeny both in the liver and in the intestine was applied (Upreti liver + intestine ontogeny). This resulted in an improved model performance (Table 6, model 1 and 3). AFE and AAFE were closer to 1 when using the Upreti liver + intestine ontogeny for all parameters except C_0_ and the AAFE for individual timepoints.

Additionally, a reduction for the age-dependent effect on the FE for the AUC and C_max_ was observed (slope in Table 6 of model 1 and 3). This is meaningful since a negative slope indicates a relatively higher prediction for younger children, whereas a slope of 0 indicates no age dependent effect. This same trend can also be seen in Figure 9. The prediction of children <14 years is worse as compared to children >14 years. This trend is especially apparent for the C_max_ prediction using standard SimCYP Upreti ontogeny.

The fold error on prediction and its observed pattern as a function of age (Figure 9) is in line with the increased dose requirements described for tacrolimus [20,21]. A PopPK analysis by Knops et al. showed an age-dependent effect on the exposure when corrected for body surface area or for body weight [21]. As such, for tacrolimus, a higher dose/m^2^ body surface area is required to reach the same exposure in children <14 years [21]. As younger children receive this higher dose/m^2^ body surface area, this translates into a consistent overprediction when simulating younger children with the Emoto or Upreti ontogeny. Based on the observed results of an improved prediction when using the Upreti liver + intestine ontogeny, it can be assumed that the cause for this higher dose requirement might be due to an increased CYP3A4 metabolism in the intestine.

This postulated ontogeny Upreti liver + intestine is in contrast to the current SimCYP default intestinal ontogeny assumptions. In SimCYP, the standard intestinal CYP3A4 ontogeny when using Upreti ontogeny profile in the liver (profile 2) is flat (green dotted line in Figure 3). When using the Salem or Emoto ontogeny for liver CYP3A4 levels are even lower in young children (Johnson ontogeny, orange dotted line in Figure 3). Measured data on CYP3A4 expression levels in children to substantiate such hypothesis are very rare. Kiss, Johnson, and Chen et al. observed this lower protein expression of CYP3A4 in young children [40,44,45]. In contrast, Fakhoury et al. observed a ~two-fold villin mRNA-corrected increase in CYP3A4 mRNA expression in the duodenum in children from 1–6 yo compared to children >6 yo [46]. The highest predicted mRNA expression was around the age of 3, which would be in line with an ontogeny profile which shows an increased expression for younger children. Given these limited and somewhat contradictory literature data, we conclude that the ontogeny of intestinal CYP3A4 should be further investigated. Based on the observed increased dose requirements of tacrolimus and this PBPK analysis, we propose that a similar Upreti ontogeny in liver and gut could be considered.

Additionally, a similar situation of disconnected in silico performance and in vivo expression measurements is present for hepatic CYP3A4 expression. As discussed by Johnson et al., two major options for liver CYP3A4 expression (Salem ontogeny and Upreti ontogeny) are available and are also visualised in Figure 3. Both ontogeny profiles were developed based on exposure profiles of intravenous drugs, but Salem et al. used midazolam exposure while Upreti et al. used sufentanil exposure to deduce ontogeny profiles. Johnson et al. showed the better performance of the Upreti ontogeny in predicting midazolam exposure, though, as also discussed in their paper, their in vivo measured CYP3A4 levels correlate better with the Salem ontogeny [43].

A third parameter which might influence the increased metabolism is the contribution of P-gp. Potentially, P-gp mediated efflux of tacrolimus back into the intestinal lumen may reduce saturation of CYP3A4 and eventually increase intestinal metabolism [47]. However, as also discussed by Jogiraju et al., kinetics for P-gp transport of tacrolimus are still lacking [48] and this transporter could not be included in the model. Additionally, it should be noted that a fairly stable, age-independent expression for P-gp in the intestine has been reported, making it less likely that P-gp is responsible for the observed age-dependent effect on tacrolimus disposition [49].

##### Exploration of the Impact of Dissolution Rate and Bile Salt Levels on Paediatric Absorption

To investigate the impact of the dissolution rate on paediatric absorption of tacrolimus, the performance of the DLM was compared to the input of biorelevant in vitro dissolution data. The predictive error using the custom dissolution in vitro profiles slightly increased except for the C_0_ prediction. However, the effects are minimal; they are visualized in Appendix A.

Secondly, as a lower bile salt secretion is observed for younger children [50,51], the impact of a decreased bile salt concentration on tacrolimus exposure was explored using the DLM. As can be seen in Appendix A, using the ½ FaSSIF and FaSSIF solubility, the prediction of PK parameters was mostly unaffected. An additional sensitivity analysis on the K_m:w_ over the 5–95% confidence interval (3.6–5.55) showed low to no sensitivity of the model performance.

##### Sensitivity Analysis of Tacrolimus Fraction Absorbed in Gastroplus

To further explore the contribution of absorption to the variability of tacrolimus exposure, and more specifically the SIFV, the Gastroplus model was used (Figure 2, step 8). In line with the SimCYP results, in the simulated fraction absorbed when assuming the solubility of the amorphous tacrolimus formulation, a low sensitivity to changes in the SIFV was predicted. Fractions absorbed ranged between 98.65% (for the 10.5 mg dose and 4.0 mL SIFV) and 99.82% (for the 0.625 mg dose and 225.5 mL SIFV) even though tacrolimus is a BCS class 2 compound.

When using crystalline tacrolimus solubility and repeating the same sensitivity analysis on the SIFV, a higher sensitivity and lower fraction absorbed were observed. Here the fraction absorbed varied between 44% and 52% for the different dose/volume conditions. This lower fraction absorbed originates from a slower absorption due to the lower solubility. Based on the simulated results, the unabsorbed fraction is excreted as unchanged tacrolimus.

Lastly, precipitation time was predicted to have a minor effect on the fraction absorbed of the amorphous formulation (Figure 2, step 9). Even in the simulated worst-case scenario (very fast precipitation), the fraction absorbed remained practically unchanged. This probably relates to the high permeability which allows rapid absorption of the dissolved fraction.

Overall, tacrolimus dissolution appears not to cause a substantial variation in oral exposure, which can be explained by the formulation properties, pharmacodynamic properties, and the paediatric physiology. First of all, tacrolimus is a potent drug, requiring a low dose. Additionally, even though tacrolimus is a BCS class 2 compound due to its low crystalline solubility, the commercial formulation is based on an amorphous solid dispersion, resulting in significantly higher intestinal concentrations. As such, the tacrolimus formulation behaves more as a BCS class 1 compound with rapid and complete dissolution combined with a good permeability.

Finally, even though PopPK analysis shows a trend in increased weight-corrected dose requirements for tacrolimus [20,21], the weight-corrected dose for children tends to decrease faster than the measured gastrointestinal fluid volumes as highlighted by the dose/volume ratio in Table 1. As such, no solubility issues arise for younger children.

##### Exploring the Impact of Distribution on Tacrolimus Exposure Using Sensitivity Analysis

Red blood cell partitioning

Distribution was also explored as a potential source of variability. As previously discussed, the observed high blood:plasma partitioning observed for tacrolimus seems to be driven by the binding FKBP inside the red blood cells and for which kinetic studies show a high importance [42]. Since expression levels for FKBP and ontogeny profiles are unavailable, a full mechanistic modelling of this binding is not possible. However, to explore potential age-related differences, a lower B_max_ for children was hypothesized. From a physiological perspective, this would correspond to a lower amount of FKBP in red blood cells, leading to a reduced tacrolimus concentration in the red blood cell with possibly lower C_max_- and C_0_-values. To represent an assumed ontogeny profile for FKBP-binding proteins, the B_max_ for 3 yo patients was fitted and the interpolated B_max_ values as shown in Figure 4 were used. Simulations using the model including this assumed ontogeny for the B_max_ of red blood cell partitioning showed an improvement in C_max_ prediction (AAFE from 1.52 to 1.38 (Table 6, model 4)) but a decreased performance in C_0_ prediction (AAFE from 1.46 to 1.79 (Table 6)). Other PK parameters were predicted with roughly the same error (AUC) or slightly worse or better (k_e_ and individual timepoints). Figure 10 shows the prediction of the selected reference data using both the B_max_ fitted model and the standard DLM with the Upreti ontogeny both in the liver and intestines.

2.Fraction Unbound in Plasma

A higher fraction unbound in plasma has been suggested in children [4,20], but our model showed low sensitivity to this. A two-fold increase in the fraction unbound led to a median C_max_ reduction from 39 ng/mL to 34 ng/mL for a 3-year-old patient and the effect seemed to level out with C_max_ of 32.2 ng/mL and 31.6 ng/mL for three- and four-fold increases, respectively. Based on this low sensitivity, the fraction unbound in plasma is not expected to be a substantial factor in age-dependent pharmacokinetics of tacrolimus. Most likely, this is related to tacrolimus’ high partitioning into red blood cells.

## 4. Conclusions

In conclusion, a new PBPK model for tacrolimus was developed and applied for scaling of PK to children, showing the value of PBPK modelling in specific populations. We believe that our model is the first to include both mechanistic blood:plasma partitioning and a mechanistic absorption model. The model was extensively validated with adult literature reference data in healthy subjects and kidney transplant patients, and with an in-house PK dataset obtained in paediatric kidney transplant patients. We found that the amorphous solid dispersion leads tacrolimus to behave as a BCS class 1 rather than class 2 compound. As a result, the absorption of tacrolimus did not appear a major factor in explaining the variability in oral exposure to tacrolimus in adults and paediatrics. However, the current reference data only included kidney transplant patients; model performance for children receiving different organ transplants might be interesting. Additionally, this only concerns data from a single clinical centre and as such, external validation would be valuable.

The ontogeny of CYP3A4 expression turned out to be a critical factor when extrapolating the adult model for tacrolimus to the paediatric population. The best model performance was obtained when using the Upreti ontogeny in not only the liver but also the intestines, suggesting the importance of increased hepatic and intestinal metabolism for tacrolimus exposure in younger children. As this is, to the best of our knowledge, the first paper to introduce this hypothesis, validation with different drugs would be valuable. From a distribution perspective, an age-dependent blood:plasma partitioning (B_max_-value) further improved the model performance. However, this impact is hard to substantiate from a mechanistic perspective as reference data are lacking. Finally, the model showed little sensitivity of tacrolimus PK to possible age-dependent changes in the fraction unbound in plasma.

## Figures and Tables

**Figure 2 pharmaceutics-15-02231-f002:**
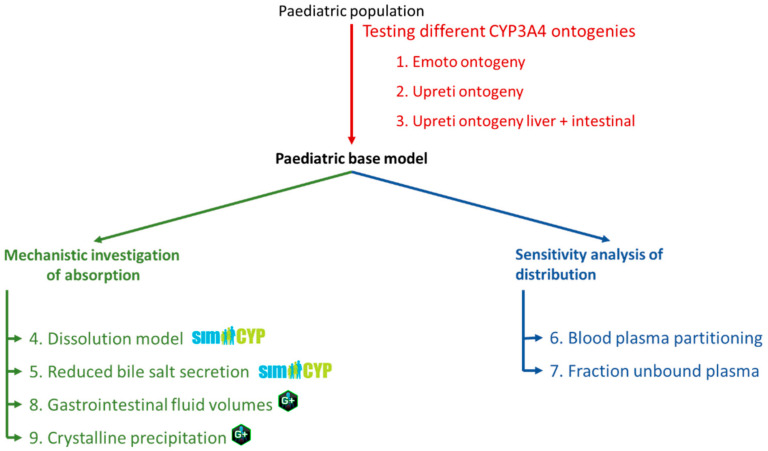
Modelling strategy used for the paediatric population.

**Figure 3 pharmaceutics-15-02231-f003:**
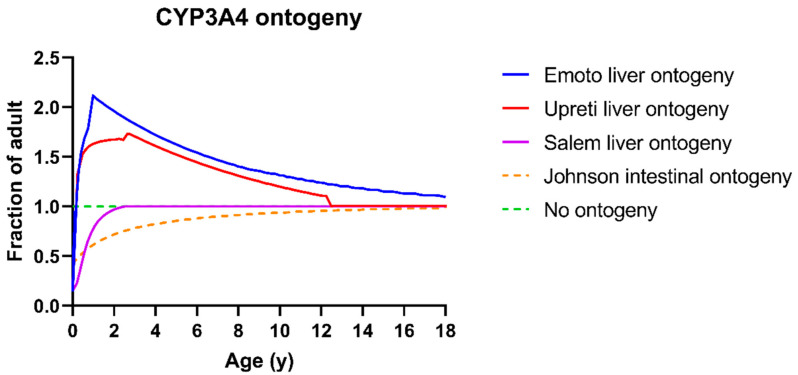
Ontogeny profiles of CYP3A4 in both the intestine and the liver visualized as the fraction of adult expression as a function of age. Blue = liver CYP3A4 ontogeny profile as published by Emoto et al., red = Simcyp Upreti liver CYP3A4 ontogeny profile, dotted green = no ontogeny, purple = Simcyp Salem liver CYP3A4 ontogeny profile, dotted orange = SimCYP Johnson intestinal CYP3A4 ontogeny profile.

**Figure 4 pharmaceutics-15-02231-f004:**
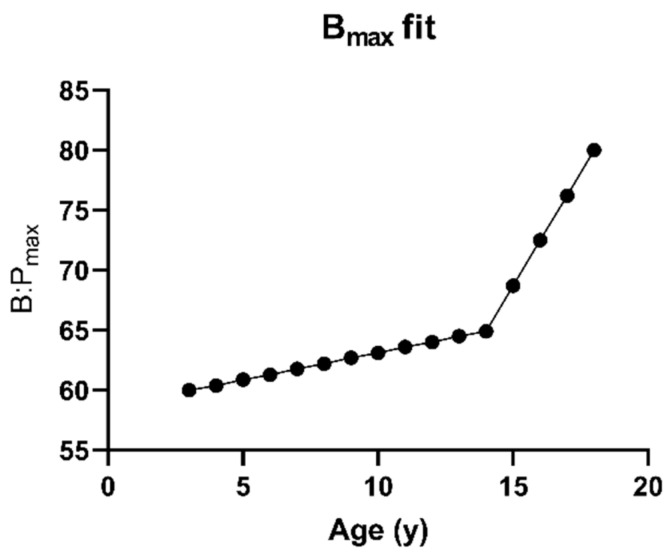
Fitted B_max_ values for blood plasma partitioning in the paediatric population.

**Figure 5 pharmaceutics-15-02231-f005:**
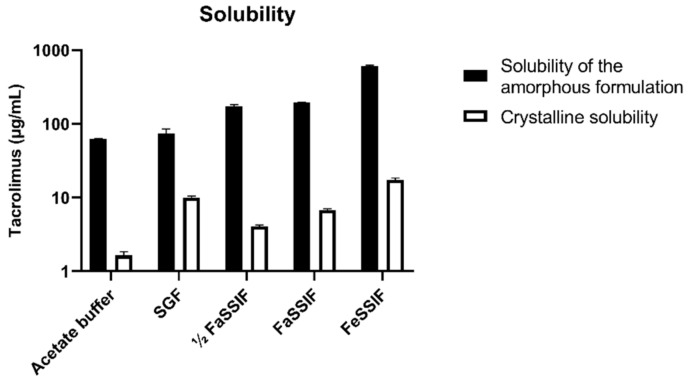
Crystalline and amorphous solubility of tacrolimus in different media. Open bars indicate the crystalline solubility; full black bars indicate the amorphous solubility, measured from the Prograf formulation.

**Figure 6 pharmaceutics-15-02231-f006:**
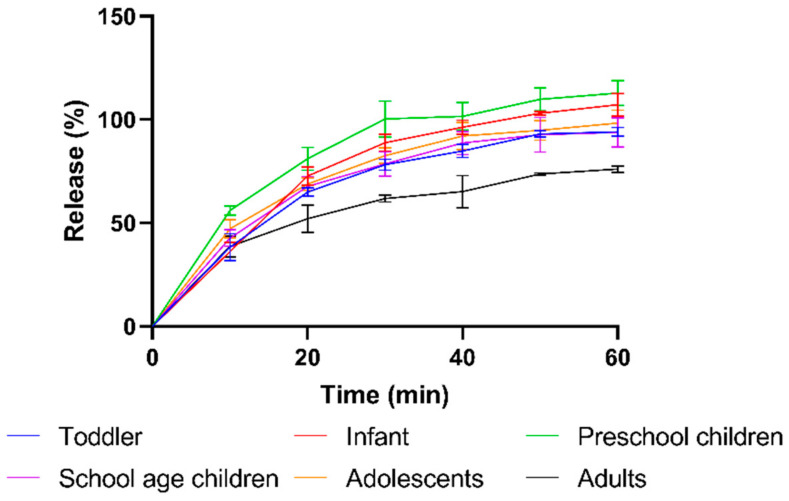
Dissolution profiles of the amorphous tacrolimus formulation Prograf using the dose-to-volume ratios representative for adult and paediatric subpopulations. Blue = Toddler, Red = Infant, Green = Preschool children, Purple = School age children, Orange = Adolescents, Black = Adults.

**Figure 7 pharmaceutics-15-02231-f007:**
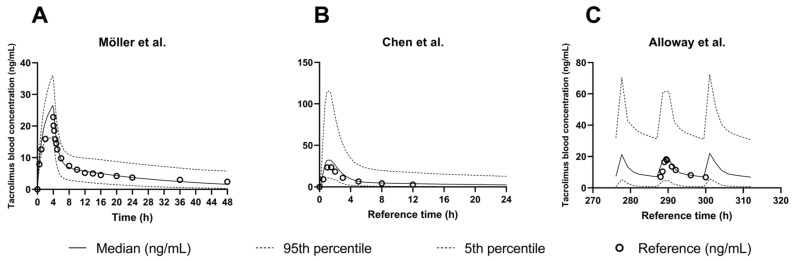
Tacrolimus exposure in adults: comparison of simulations using the adult PBPK model to selected clinical studies. (**A**) Möller et al. [3] as a reference study for a single IV administration (0.01 mg/kg). (**B**) Chen et al. [32] as a reference study for a single oral administration (0.075 mg/kg) to kidney transplant patients (**C**) Alloway et al. [34] as a reference study for tacrolimus exposure at steady state after oral administration of 2.85 mg in kidney transplant patients twice per day. In (**B**,**C**), the diffusion layer model was used to simulate dissolution. Full lines indicate the median simulated profile, dotted lines indicate the 5% and 95% percentile, and open circles represent the average reference data.

**Figure 8 pharmaceutics-15-02231-f008:**
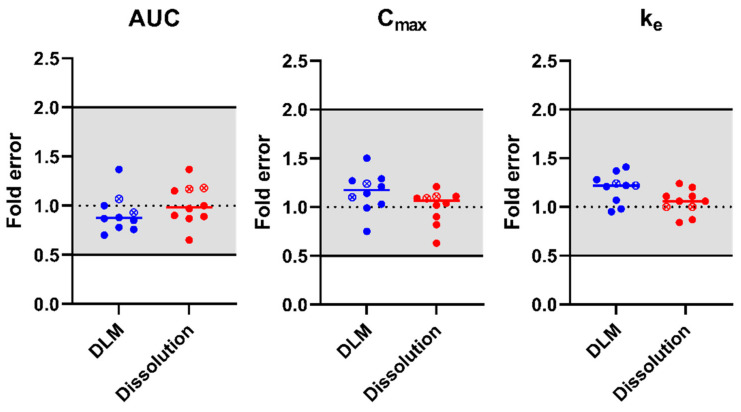
Fold error on simulated PK parameters using either the DLM or the input of an in vitro dissolution profile to simulate drug dissolution. Dotted line indicates an ideal prediction, grey coloured area indicates the 2-fold acceptance criteria, while the short full lines indicate the average fold error. Full circles are PK parameters after single dose administrations, open circles with an x inside are at steady state.

**Figure 9 pharmaceutics-15-02231-f009:**
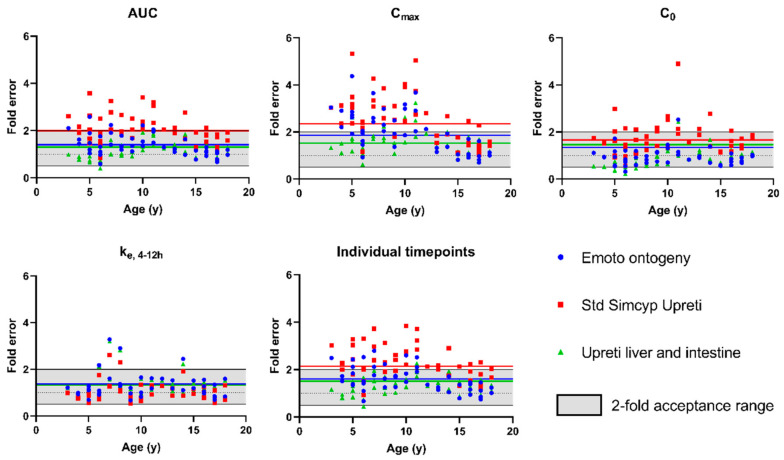
Fold error of different paediatric PBPK models for tacrolimus on simulated PK parameters as a function of age. Grey coloured area indicates the 2-fold acceptance criteria. Blue dots = simulations using the Emoto ontogeny, Red squares = simulations using the Upreti ontogeny, Green triangle = simulations using the Upreti liver + intestine ontogeny. Full line indicates the respective AAFE for each PK parameter and model. All fold errors are on concentrations at steady state.

**Figure 10 pharmaceutics-15-02231-f010:**
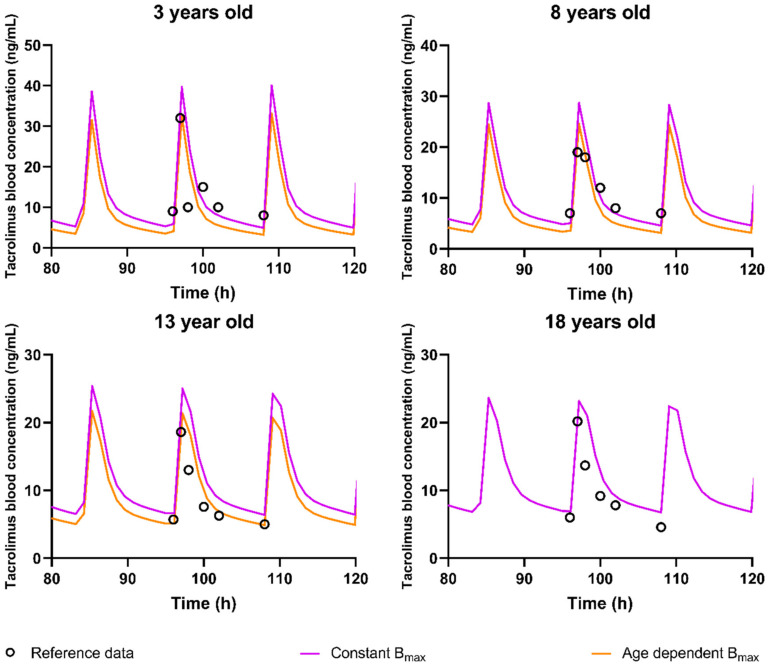
Representative simulations for different ages at steady state for the different models. Open circles indicate the reference data, Purple = simulations using the Upreti ontogeny in liver and intestines assuming constant B_max_, Orange = Simulation using the Upreti ontogeny both in liver and intestines and with the assumed age-dependent change in B_max_.

**Table 1 pharmaceutics-15-02231-t001:** Doses and volumes used in the age-dependent dissolution experiments with tacrolimus. SIFV were measured using magnetic resonance imaging (MRI) and published by Mudie et al. [22] for adults and Van der Veken et al. [23] for paediatric patients. For al (sub)populations, a 0.15 mg/kg dose was assumed.

Subpopulation	Average SIFV Reported (mL)	Volume Used In Vitro (mL)	Body Weight (kg)	Dose (mg)	Dose/Volume Ratio (mg/mL)
Infant (0.1–1 year)	22.1	25	7.3	1.09	0.04
Toddler (1–2 year)	35.1	35	10.1	1.51	0.04
Preschool child (2–5 year)	38.5	40	10.8	1.63	0.04
School-age child (6–11 year)	48.5	50	22.0	3.30	0.07
Adolescent (12–16 year)	94.0	50	53.4	8.01	0.16
Adult	43.0	50	70.0	10.50	0.21

**Table 2 pharmaceutics-15-02231-t002:** Parameters of the used UPLC-UV and UPLC-MS method for tacrolimus quantification. MP = mobile phase.

		MS Method	UV Method		
Column		Kinetex 2.6 µm XB-C18 100A 50 × 2.1 mm	Acquity UPLC BEH-C18 1.7 µm 50 mm × 2.1 mm		
Flow (mL/min)		0.6	0.8		
Column temperature (°C)		55	60		
Injection volume		5	80		
Time (min)	MP 1 (%)	MP 2 (%)	Time (min)	MP 3 (%)	MP 4 (%)
0	25	75	0	55	45
0.5	97.5	2.5	8	45	55
2	25	75	8.5	55	45
3.2	25	75	12	55	45
Detection	Tacrolimus-NH4	821.7 → 768.6	UV	210 nm	
	Tacrolimus-NH4 D5	826.6 → 773.6			

Mobile phase 1 = Acetonitrile, mobile phase 2 = Ammonium acetate 2 mM + 0.1% formic acid in water, mobile phase 3 = 90% water with 1% trifluoroacetic acid—10% acetonitrile, mobile phase 4 = 90% acetonitrile—10% water.

**Table 4 pharmaceutics-15-02231-t004:** Summary of the combinations of liver and intestinal CYP3A4 and CYP3A5 ontogeny profiles described in this paper.

Name in This Paper	Liver CYP3A4 Ontogeny	Intestinal CYP3A4 Ontogeny	Liver CYP3A5 Ontogeny	Intestinal CYP3A5 Ontogeny
Salem ontogeny	SimCYP profile 1— Salem et al.	SimCYP default— Johnson et al.	SimCYP default— no ontogeny	SimCYP default—Johnson et al.
Upreti ontogeny	SimCYP profile 2— Upreti et al.	No ontogeny		
Emoto ontogeny	Custom refitting of Salem et al. [16]	SimCYP default— Johnson et al.		
Upreti liver + intestine ontogeny	SimCYP profile 2— Upreti et al.	SimCYP profile 2— Upreti et al.		

**Table 5 pharmaceutics-15-02231-t005:** Gastroplus PBPK model input parameters.

	Adult Model	References
**Population**	Healthy Volunteers	
**Drug Parameters**		
Physicochemical properties		
Molecular weight (g/mol)	804.02	[36]
Log P_o:w_	3.26	[36]
Compound type	neutral	[36]
**Absorption**		
ACAT model		
P_eff_ (10^−4^ cm/s)	4.77	[26]
Formulation	IR Capsule	
Solubility @ pH 4.5 ABS buffer—Amorphous formulation (mg/mL) (Reference solubility)	0.06257	In-house data
SGF solubility—Amorphous formulation (mg/mL)	0.07412	In-house data
FaSSIF Solubility—Amorphous formulation (mg/mL)	0.19655	In-house data
FeSSIF Solubility—Amorphous formulation (mg/mL)	0.60547	In-house data
Solubilization ratio—Amorphous formulation	30300	Gastroplus fitted
Solubility @ pH 4.5 ABS buffer—Crystalline (mg/mL)	0.00165	In-house data
SGF solubility—Crystalline (mg/mL)	0.01003	In-house data
FaSSIF Solubility—Crystalline (mg/mL)	0.00678	In-house data
FeSSIF Solubility—Crystalline (mg/mL)	0.01746	In-house data
Solubilization ratio—Crystalline	11400	Gastroplus fitted
Mean particle radius (µm)	25	Default value in Gastroplus
Mean precipitation time (sec)	900	Default value in Gastroplus
Particle density (g/mL)	1.2	Default value in Gastroplus
**Distribution**		
Distribution model	none used	
**PSA input**		
Dose (mg)	10.5, 5, 2.5, 1.25 or 0.625	
PSA analysis setup	Input value	
Small intestine fluid volume (minimum)	0.28% filled (4.0 mL)	
Small intestine fluid volume (baseline)	3.15% filled (45.2 mL)	
Small intestine fluid volume (maximum)	15.75% filled (225.5 mL)	
Precipitation time (minimum)	0.1 s	
Precipitation time (baseline)	90 s	
Precipitation time (maximum)	900 s	

ACAT model = advanced compartmental absorption and transit model; P_eff_ = effective permeability.

**Table 6 pharmaceutics-15-02231-t006:** Performance of paediatric PBPK models for tacrolimus using different ontogeny functions. AFE = average fold error, AAFE = absolute average fold error, Slope = slope of the fold errors as a function of age. Lowest AAFE values are marked in bold.

Model #	Ontogeny Profile and Fit		AUC	C_max_ (ng/mL)	C_0_ (ng/mL)	k_e_ (ng/mL/h)	Individual Timepoints
1	Liver: Emoto ontogeny	AFE	1.30	1.77	0.88	1.23	1.28
	Intestine: Johnson ontogeny	AAFE	1.40	1.86	**1.34**	1.37	1.61
	Extra fit: n/a	Slope	−0.05	−0.13	−0.01	−0.02	−0.06
2	Liver: Upreti ontogeny	AFE	1.98	2.34	1.62	0.98	1.87
	Intestine: No ontogeny	AAFE	1.99	2.34	1.66	**1.32**	2.14
	Extra fit: n/a	Slope	−0.04	−0.12	0.00	−0.01	−0.05
3	Liver: Upreti ontogeny	AFE	1.10	1.44	0.78	1.18	1.11
	Intestine: Upreti ontogeny	AAFE	**1.30**	1.52	1.46	1.34	**1.51**
	Extra fit: n/a	Slope	0.01	−0.03	0.03	−0.02	0.01
4	Liver: Upreti ontogeny	AFE	0.90	1.24	0.59	1.26	0.94
	Intestine: Upreti ontogeny	AAFE	**1.30**	**1.38**	1.79	1.40	1.58
	Extra fit: B_max_ fit	Slope	0.03	−0.01	0.04	−0.03	0.02

## Data Availability

Data is contained within the article and Appendix A.

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
