# Peer review of "Investigating Tacrolimus Disposition in Paediatric Patients with a Physiologically Based Pharmacokinetic Model Incorporating CYP3A4 Ontogeny, Mechanistic Absorption and Red Blood Cell Binding"

_pharmaceutics, 2023, doi:10.3390/pharmaceutics15092231_

Round 1

Reviewer 1 Report

The paper introduces an approach that combines in vitro dissolution data and PBPK modeling to predict tacrolimus pharmacokinetic (PK) profiles in children, taking into account CYP3A4 ontogeny, mechanistic absorption, and RBC binding. The utilization of such approaches for PK-studies in children is essential considering that they represent a special population with limited clinical data available. The paper is interesting and well-presented, and it could be further proceeded for publication.

Author Response

Manuscript pharmaceutics-2542411

Investigating tacrolimus disposition in paediatric patients with a PBPK model incorporating CYP3A4 ontogeny, mechanistic absorption and red blood cell binding

General comments

__________________________________________________________________________________

We would like to thank the reviewers for their kind words and suggestions. Changes have been made to the initial manuscript based on the comments made by the reviewers. Furthermore, we tried to reply to the reviewers’ comments as clearly as possible and hope these replies are found to be satisfactory.

Point-to-point reply

__________________________________________________________________________________

The following is a point-to-point reply to the reviewers’ comments. The initial remarks of the reviewers are presented in bold. Key changes to the manuscript are presented in red. In the revised version of the manuscript, changes are tracked by track changes in red. References to page and line numbers apply to the revised manuscript.

Comments and Suggestions for Authors

__________________________________________________________________________________

The paper introduces an approach that combines in vitro dissolution data and PBPK modeling to predict tacrolimus pharmacokinetic (PK) profiles in children, taking into account CYP3A4 ontogeny, mechanistic absorption, and RBC binding. The utilization of such approaches for PK-studies in children is essential considering that they represent a special population with limited clinical data available. The paper is interesting and well-presented, and it could be further proceeded for publication.

Reviewer 2 Report

the paper is very interesting and the work done was carried out with extreme competency. The manuscript is well organized and written and it is very clear to follow the procedures and the discussion. Materials and methods are appropriately detailed. Accordingly, I recommend the publication of the paper. I have only very minor comments that should be interesting to address:

-I believe that in the conclusion a short paragraph enclosing the potentiality of PK modeling should be reported as well as the application to the described method to other drugs. I suggest to report also limitation of the study.

English is fine with minor typo errors

Author Response

Manuscript pharmaceutics-2542411

Investigating tacrolimus disposition in paediatric patients with a PBPK model incorporating CYP3A4 ontogeny, mechanistic absorption and red blood cell binding

General comments

__________________________________________________________________________________

We would like to thank the reviewers for their kind words and suggestions. Changes have been made to the initial manuscript based on the comments made by the reviewers. Furthermore, we tried to reply to the reviewers’ comments as clearly as possible and hope these replies are found to be satisfactory.

Point-to-point reply

__________________________________________________________________________________

The following is a point-to-point reply to the reviewers’ comments. The initial remarks of the reviewers are presented in bold. Key changes to the manuscript are presented in red. In the revised version of the manuscript, changes are tracked by track changes in red. References to page and line numbers apply to the revised manuscript.

Comments and Suggestions for Authors

__________________________________________________________________________________

the paper is very interesting and the work done was carried out with extreme competency. The manuscript is well organized and written and it is very clear to follow the procedures and the discussion. Materials and methods are appropriately detailed. Accordingly, I recommend the publication of the paper. I have only very minor comments that should be interesting to address:

  1. I believe that in the conclusion a short paragraph enclosing the potentiality of PK modeling should be reported as well as the application to the described method to other drugs. I suggest to report also limitation of the study.

To clarify these aspects, the following lines were added to the conclusion.

  • Line 630: “, showing the value of PBPK modelling in specific populations”
  • Line 639-641: “Additionally, this only concerns data from a single clinical centre and as such, external validation would be valuable.“
  • Line 646-647: “As this is, to the best of our knowledge, the first paper to introduce this hypothesis, validation with different drugs would be valuable.”

Reviewer 3 Report

The manuscript from Van der Veken et al. describes the development of a model to predict tacrolimus pharmacokinetics in adult and pediatric patient populations. This model, which took into account several physiological factors, including red blood cell distribution, was then validated by comparison with real samples.

The manuscript is of great interest because it combines in vitro experimental data with previously made clinical observations. The results obtained are convincing, especially with regard to adult patients. As far as children are concerned, some significant discrepancies are observed between the proposed models and patient data. My main doubt about this manuscript concerns precisely this last point. Indeed, I believe that a cohort of 18 patients between the ages of 3 and 18 may be too small to provide sufficient data for effective validation. The physiological variations that occur in children in the age range taken into consideration are such as to require a large sampling. Furthermore, even variables such as gender make the picture even more complex. In light of these considerations, it would be appropriate for the authors to provide a detailed picture of the 18 pediatric patients considered and, possibly, to expand this cohort.

Minor points Some information relating to the analytical aspects should be integrated: which instruments were used to carry out the LC-MS/MS and LC/UV analyses? what are the validated performances for the two analytical methods?

Author Response

Manuscript pharmaceutics-2542411

Investigating tacrolimus disposition in paediatric patients with a PBPK model incorporating CYP3A4 ontogeny, mechanistic absorption and red blood cell binding

General comments

__________________________________________________________________________________

We would like to thank the reviewers for their kind words and suggestions. Changes have been made to the initial manuscript based on the comments made by the reviewers. Furthermore, we tried to reply to the reviewers’ comments as clearly as possible and hope these replies are found to be satisfactory.

Point-to-point reply

__________________________________________________________________________________

The following is a point-to-point reply to the reviewers’ comments. The initial remarks of the reviewers are presented in bold. Key changes to the manuscript are presented in red. In the revised version of the manuscript, changes are tracked by track changes in red. References to page and line numbers apply to the revised manuscript.

Comments and Suggestions for Authors

__________________________________________________________________________________

The manuscript from Van der Veken et al. describes the development of a model to predict tacrolimus pharmacokinetics in adult and pediatric patient populations. This model, which took into account several physiological factors, including red blood cell distribution, was then validated by comparison with real samples.

  1. The manuscript is of great interest because it combines in vitro experimental data with previously made clinical observations. The results obtained are convincing, especially with regard to adult patients. As far as children are concerned, some significant discrepancies are observed between the proposed models and patient data. My main doubt about this manuscript concerns precisely this last point. Indeed, I believe that a cohort of 18 patients between the ages of 3 and 18 may be too small to provide sufficient data for effective validation. The physiological variations that occur in children in the age range taken into consideration are such as to require a large sampling. Furthermore, even variables such as gender make the picture even more complex. In light of these considerations, it would be appropriate for the authors to provide a detailed picture of the 18 pediatric patients considered and, possibly, to expand this cohort.

The used dataset contains data collected over 20 years. While more data is available, these children receive other co-medications of which drug-drug interactions are expected (e.g. antiviral medications with CYP3A4 inhibition). As such this would introduce other complexities.

While collecting more data is not possible in a short time frame, we do agree there are limitations to the used dataset and added the following statements to the conclusion.

  • Line 639-641: “Additionally, this only concerns data from a single clinical centre and as such, external validation would be valuable.“
  • Line 646-647: “As this is, to the best of our knowledge, the first paper to introduce this hypothesis, validation with different drugs would be valuable.”

 With regards to the publication of clinical characteristics of the individual patients, as this concerns clinical, sensitive data, we are unable to publish further details.

  1. Minor points Some information relating to the analytical aspects should be integrated: which instruments were used to carry out the LC-MS/MS and LC/UV analyses? what are the validated performances for the two analytical methods?

Extra information regarding the used analytical method (linearity etc.) is available in the supplementary data.
